# Attenuation of Free Fatty Acid (FFA)-Induced Skeletal Muscle Cell Insulin Resistance by Resveratrol is Linked to Activation of AMPK and Inhibition of mTOR and p70 S6K

**DOI:** 10.3390/ijms21144900

**Published:** 2020-07-11

**Authors:** Danja J. Den Hartogh, Filip Vlavcheski, Adria Giacca, Evangelia Tsiani

**Affiliations:** 1Department of Health Sciences, Brock University, St. Catharines, ON L2S 3A1, Canada; dd11qv@brocku.ca (D.J.D.H.); fvlavcheski@brocku.ca (F.V.); 2Centre for Bone and Muscle Health, Brock University, St. Catharines, ON L2S 3A1, Canada; 3Department of Physiology, University of Toronto, Toronto, ON M5S 1A8, Canada; adria.giacca@utoronto.ca; 4Department of Medicine, University of Toronto, Toronto, ON M5S 1A8, Canada; 5Institute of Medical Sciences, University of Toronto, Toronto, ON M5S 1A8, Canada; 6Banting and Best Diabetes Centre, University of Toronto, Toronto, ON M5G 2C4, Canada

**Keywords:** resveratrol, free fatty acid, palmitate, insulin resistance, IRS-1, mTOR, p70 S6K, AMPK, GLUT4, Akt

## Abstract

Insulin resistance, a main characteristic of type 2 diabetes mellitus (T2DM), is linked to obesity and excessive levels of plasma free fatty acids (FFA). Studies indicated that significantly elevated levels of FFAs lead to skeletal muscle insulin resistance, by dysregulating the steps in the insulin signaling cascade. The polyphenol resveratrol (RSV) was shown to have antidiabetic properties but the exact mechanism(s) involved are not clearly understood. In the present study, we examined the effect of RSV on FFA-induced insulin resistance in skeletal muscle cells in vitro and investigated the mechanisms involved. Parental and GLUT4myc-overexpressing L6 rat skeletal myotubes were used. [^3^H]2-deoxyglucose (2DG) uptake was measured, and total and phosphorylated levels of specific proteins were examined by immunoblotting. Exposure of L6 cells to FFA palmitate decreased the insulin-stimulated glucose uptake, indicating insulin resistance. Palmitate increased ser^307^ (131% ± 1.84% of control, *p* < 0.001) and ser^636/639^ (148% ± 10.1% of control, *p* < 0.01) phosphorylation of IRS-1, and increased the phosphorylation levels of mTOR (174% ± 15.4% of control, *p* < 0.01) and p70 S6K (162% ± 20.2% of control, *p* < 0.05). Treatment with RSV completely abolished these palmitate-induced responses. In addition, RSV increased the activation of AMPK and restored the insulin-mediated increase in (a) plasma membrane GLUT4 glucose transporter levels and (b) glucose uptake. These data suggest that RSV has the potential to counteract the FFA-induced muscle insulin resistance.

## 1. Introduction

Skeletal muscle tissue accounts for nearly 75–80% of the postprandial insulin-mediated glucose uptake, thereby it plays a predominant role in maintaining glucose homeostasis. In skeletal muscle cells, insulin increases glucose uptake by binding to its receptor, leading to increase in its tyrosine kinase activity and downstream phosphorylation of the insulin receptor substrate-1 (IRS-1), and activation of phosphatidylinositol-3 kinase (PI3K) and protein kinase B/Akt [1,2]. The PI3K/Akt pathway plays a pivotal role in insulin-stimulated glucose transporter 4 (GLUT4) translocation to the plasma membrane [1,2]. Impairments in the IRS-1-PI3K-Akt signaling cascade results in insulin resistance and type 2 diabetes mellitus (T2DM) [3,4]. Muscle insulin resistance is a key contributor to reduced glucose tolerance and T2DM. Elevated free fatty acid (FFA) levels, often seen in obesity, are closely linked to insulin resistance and are a major risk factor for the development of T2DM [5,6]. In vitro evidence showed that exposure of muscle cells to FFA palmitate resulted in insulin resistance [7,8]. Furthermore, in vivo animal studies found that increased plasma lipid levels achieved by a high fat diet or lipid infusion resulted in muscle insulin resistance [9,10]. Evidence shows that FFAs significantly impair the insulin signaling by increasing serine phosphorylation of IRS-1 [11,12], through activation/phosphorylation of protein kinases, including mechanistic target of rapamycin (mTOR) [13,14], ribosomal protein S6 kinase (p70 S6K) [15,16], c-Jun N-terminal kinase (JNK) [12], protein kinase C (PKC) [15], glycogen synthase kinase 3 (GSK3) [17], and inhibitory kappa B (IκB) kinase (IKK) [7,18].

The energy sensor adenosine monophosphate (AMP)-activated protein kinase (AMPK) is activated by energy depletion or an increased AMP/ATP ratio and phosphorylation by its upstream kinases calmodulin-dependent protein kinase (CaMKKs), liver kinase B1 (LKB1), and transforming growth factor-β (TGF-β)-activated kinase 1 (TAK1) [19,20]. Phosphorylation/activation of AMPK in muscle cells was observed in response to contraction/exercise and exposure to chemicals that inhibit mitochondrial adenosine triphosphate (ATP) production, such as 2,4-dinitrophenol (DNP) [21]. Furthermore, there are several compounds that activate AMPK, including metformin [22], thiazolidineones [23], salicylate [24], and polyphenols, such as quercetin [25], berberine [26], and resveratrol [27,28,29]. Recently, AMPK activators have been considered to be a novel and attractive prophylactic or treatment strategy against insulin resistance and T2DM [30].

Resveratrol (RSV) (3,5,40-trihydroxy-trans-stillbene) is a naturally derived polyphenol abundantly found in the skin of grapes and red wine and, based on the common structure of two phenyl moieties connected by a two-carbon methylene bridge, belongs to the family of stilbenes [31]. Resveratrol was shown to exhibit diverse health benefits, especially in terms of cardiovascular diseases, cancer, neurological disorders, and diabetes. In vitro studies have demonstrated that RSV increases glucose uptake in the L6 and C2C12 skeletal muscle cells, to levels comparable to insulin and metformin [28,32]. Furthermore, in vivo animal studies showed that administration of RSV significantly improved insulin sensitivity, and increased glucose tolerance and mitochondrial biogenesis in mice fed a high-fat diet [33]. These effects of RSV were largely abolished in AMPK-α1 or -α2 knockout mice, indicating that AMPK plays a pivotal role in mediating the effects of RSV [34]. Additionally, we recently demonstrated that RSV ameliorated the high insulin-induced skeletal muscle cell insulin resistance, and that these effects were linked to AMPK activation/phosphorylation [29].

According to the International Diabetes Federation (IDF), approximately 390 million people are currently affected by T2DM, with nearly 4.5 million annual worldwide deaths reported in 2019, resulting in an enormous economic burden to healthcare systems [35]. The majority of T2DM-related deaths are due to serious complications and lack of adequate treatment/management. Uncontrolled T2DM might lead to cardiovascular, kidney, nerve, liver, and eye damage [36]. Despite recent improvements in T2DM therapy, more cost-effective and efficient treatments with less side-effects than the current medications are still needed. Plant-derived chemicals have attracted attention to be used for the treatment and prevention of insulin resistance and T2DM. Although numerous studies have examined the antidiabetic effects of RSV, the cellular mechanisms of its action to counteract insulin resistance are not fully elucidated. The focus of this study was to investigate the mechanism of action of RSV in palmitate-induced muscle insulin-resistant myotubes.

## 2. Results

### 2.1. Resveratrol Abolishes the Palmitate-Induced Ser307 and Ser636/639 Phosphorylation of IRS-1

Previous studies performed in L6 muscle cells in vitro [8,29,37] and rat muscle tissue in vivo [38] indicated that increased serine (ser^307^ and ser^636/639^) phosphorylation of IRS-1 leads to impairments within the insulin signaling cascade, resulting in insulin resistance. Hence, we examined the effects of palmitate and RSV on serine phosphorylation and expression of IRS-1. Exposure of L6 myotubes to palmitate (P) (0.4 mM, 16 h) increased the ser^307^ and ser^636/639^ phosphorylation of IRS-1 (P: 131% ± 1.8% and 148% ± 10% of control, respectively, *p* < 0.01; Figure 1). Importantly, treatment with RSV (25 µM) completely abolished this palmitate-induced phosphorylation of IRS-1 (RSV+P: 92% ± 7.4% and 102% ± 6.1% of control, *p* < 0.05; Figure 1A,B). The total levels of IRS-1 were not significantly changed by any treatment (P: 96% ± 2.9%, RSV: 98% ± 0.9%, RSV+P: 99% ± 3.5% of control; Figure 1).

### 2.2. Resveratrol Attenuates the Palmitate-Induced Phosphorylation/Activation of mTOR and p70 S6K

Increased activation of muscle mTOR by nutrient overload was shown to lead to increased serine phosphorylation of IRS-1, impaired insulin signaling, and induction of insulin resistance [39,40]. The effect of palmitate on muscle cell mTOR phosphorylation/activation was measured next. Exposure of myotubes to palmitate, significantly increased the phosphorylation of mTOR (P: 174% ± 15.4% of control, *p* < 0.01; Figure 2A,B), and this response was abolished in the presence of RSV (RSV+P: 121% ± 7.44% of control, *p* < 0.05, Figure 2A,B). The total mTOR levels were unaffected by any treatment (P: 113% ± 5.8%, RSV: 130% ± 2.34%, RSV+P: 110% ± 0.55% of control; Figure 2A,B). The data expressed as the ratio of phosphorylated mTOR to total mTOR are shown in Figure 2C. The increase in the ratio of phosphorylated to total mTOR seen with palmitate, was attenuated by RSV treatment (P: 154% ± 10.08%, RSV+P: 109% ± 6.6% of control, *p* < 0.05, compared to P; Figure 2C). It is important to note that we utilized the same PVDF membrane, cut it horizontally in three parts according to the protein ladder/molecular-weight size marker, and blotted the upper part for mTOR, the middle for p70 S6K and the lower part for β-actin. For this reason, we included the same β-actin representative blot in the figures for mTOR and p70 S6K.

Similarly, p70 S6K, the downstream effector of mTOR, is implicated in the serine phosphorylation of IRS-1, leading to impaired insulin signaling and insulin resistance [41]. Exposure of the cells to palmitate increased the phosphorylation of p70 S6K (P: 162% ± 20.2% of control, *p* < 0.05; Figure 3). Crucially, this effect of palmitate was abolished by RSV (RSV+P: 54% ± 11.8% of control, *p* < 0.01; Figure 3). The total levels of p70 S6K were not changed by any treatment. In addition, treatment of myotubes with RSV alone did not affect p70 S6K phosphorylation or expression (Figure 3A,B). The data expressed as the ratio of phosphorylated p70 S6K to total p70 S6K are shown in Figure 3C. The increase in the ratio of phosphorylated p70 S6K to total p70 S6K seen with palmitate was attenuated by RSV treatment (P: 166% ± 7.08%, RSV+P: 61.6% ± 11.11% of control, *p* < 0.001, compared to P; Figure 3C).

### 2.3. Resveratrol Increases AMPK Phosphorylation/Activation in the Presence of Palmitate

Previous studies by our group and others [28,42,43] showed that RSV acutely (15 min to 2 h) phosphorylates/activates AMPK in L6 skeletal muscle cells. In the present study, we examined the effect of RSV (16 h) on AMPK in the absence or presence of palmitate. Treatment of myotubes with RSV alone resulted in a significant increase in AMPK phosphorylation (RSV: 227% ± 22.0% of control, *p* < 0.01; Figure 4A). Treatment with palmitate alone did not have a significant effect on AMPK phosphorylation. RSV in the presence of palmitate significantly increased the ratio of phosphorylated AMPK to total AMPK (RSV+P: 193% ± 2.18% of control, *p* < 0.01; Figure 4C).

### 2.4. Resveratrol Restores the Insulin-Stimulated Glucose Uptake in Palmitate-Treated Muscle Cells

Acute exposure of L6 myotubes to insulin (I; 100 nM, 30 min) resulted in a significant increase in glucose uptake (175% ± 4.76% of basal control, *p* < 0.01; Figure 5). Treatment with palmitate (P; 0.4 mM, 16 h) had no effect on basal glucose uptake. However, palmitate significantly reduced the insulin-stimulated glucose uptake (P+I; 116 ± 2.76% of basal control, *p* < 0.001; Figure 5). Treatment with RSV alone (25 µM, 16 h) increased the basal glucose uptake (RSV: 118% ± 4.38% of control; Figure 6). Importantly, the presence of RSV in palmitate-treated cells increased the insulin-stimulated glucose uptake (RSV+P+I: 138% ± 5.55% of basal control, *p* < 0.05; Figure 5). These data indicate that the negative effect of palmitate on insulin-stimulated glucose uptake was attenuated in the presence of RSV.

### 2.5. Resveratrol Restores the Insulin-Stimulated GLUT4 Translocation in Palmitate-Treated Muscle Cells

We examined the effects of RSV on GLUT4 using L6 cells that overexpressed a myc-labelled GLUT4 glucose transporter. Acute stimulation of GLUT4myc-overexpressing L6 cells with insulin (100 nM, 30 min) significantly increased the GLUT4 plasma membrane levels (I: 189% ± 8.9% of control, *p* < 0.001), indicating translocation from an intracellular storage site to the plasma membrane. Exposure of the cells to palmitate alone (0.4 mM, 16 h) had no effect on the GLUT4 plasma membrane levels (P: 107% ± 3.91% of control; Figure 6). Palmitate attenuated the acute insulin-stimulated increase in the GLUT4 plasma membrane levels, indicating insulin resistance (P+I: 135% ± 10.05% of control, *p* < 0.05; Figure 6). Treatment with RSV alone significantly increased the basal GLUT4 plasma membrane levels (RSV: 144% ± 14.1% of control; Figure 6) and more importantly, the insulin-stimulated GLUT4 translocation was restored in the presence of RSV (RSV+P+I: 233% ± 1.8% of control, *p* < 0.01; Figure 6).

### 2.6. Resveratrol Restores the Insulin-Stimulated Akt Phosphorylation in Palmitate-Treated Muscle Cells

In addition, we examined the effects of palmitate and RSV on insulin-stimulated Akt phosphorylation/activation and expression. The insulin-stimulated increase in glucose uptake and GLUT4 glucose transporter translocation from an intracellular storage site to plasma membrane in muscle cells is mediated by Akt activation [1,2]. Phosphorylation of Akt on ser^473^ residue is an established indicator of its activation [2] and we used a specific antibody that recognized the phosphorylation of this residue. Exposure of L6 myotubes to insulin alone resulted in a significant increase in Akt phosphorylation/activation (I: 679% ± 91.8% of control, *p* < 0.01; Figure 7A,B). Treatment of the cells with palmitate (0.4 mM, 16 h) attenuated the insulin-stimulated Akt phosphorylation (P+I: 159% ± 22.9% of control, *p* < 0.05; Figure 7A,B), indicating insulin resistance. Importantly, in the presence of RSV and palmitate, the insulin-stimulated Akt phosphorylation/activation was restored to levels seen with insulin alone (RSV+P+I: 646% ± 101.6% of control, *p* < 0.05; Figure 7). The total levels of Akt were unchanged by any treatment. The data expressed as the ratio of phosphorylated Akt to total Akt are shown in Figure 7C. Treatment with palmitate abolished the insulin response and in the presence of RSV this response was restored (ratio of phosphorylated Akt to total Akt: P+I: 109% ± 15.7%, RSV+P+I: 486% ± 76.5% of control, *p* = 0.017, compared to P+I; Figure 7C).

## 3. Discussion

Elevated FFAs, as often seen in obesity, are tightly linked to insulin resistance and play a major role in the development of T2DM, a disease currently on the rise. The search for plant-derived compounds that have the potential to counteract insulin resistance is a focus of numerous research groups around the world, including ours, and such compounds might provide more effective treatment against insulin resistance and T2DM.

Skeletal muscle is a primary insulin target tissue accounting for the majority of the postprandially insulin-mediated glucose uptake and, therefore, muscle tissue plays a paramount role in maintaining glucose homeostasis. Impairments in insulin-mediated muscle glucose uptake lead to reduced glucose tolerance and T2DM. Insulin resistance and T2DM are strongly associated with elevated plasma FFAs and obesity [37,44,45]. In vitro evidence showed that exposure of muscle cells to saturated free fatty acids, such as palmitate, results in insulin resistance [7]. Furthermore, in vivo animal studies showed that administration of high-fat diet or lipid infusion results in muscle insulin resistance [9,46].

In this study, exposure of L6 myotubes to palmitate for 16 h, to mimic the elevated FFA as often seen in obesity in vivo, resulted in significant increase in ser^307^ and ser^636/639^ phosphorylation of IRS-1 (Figure 1 and Figure 8). These findings are in agreement with other in vitro studies, showing increased ser^307^ and ser^636/639^ phosphorylation of IRS-1 in L6 skeletal muscle cells [8,47], in response to exposure to palmitate. Additionally, our data are in agreement with in vivo studies demonstrating increased serine phosphorylation of IRS-1 in muscle tissue from animals fed a high-fat diet [48,49] or subjected to lipid infusion [46]. Increased serine phosphorylation of IRS-1 negatively impacted the insulin signaling cascade by reducing the tyrosine phosphorylation of IRS-1 [11] and its PI3K association/recruitment [50]. This impairment in the insulin cascade ultimately led to a reduced downstream PI3K-Akt signaling and diminished glucose uptake [38]. Most importantly, our data indicated that treatment with RSV prevented the palmitate-induced serine phosphorylation of IRS-1. Similar to our data, treatment of the L6 muscle cells with metformin, a drug used clinically in the management of blood glucose levels in T2DM, increased tyrosine phosphorylation of IRS-1, phosphorylation of Akt, and restored the insulin-stimulated glucose uptake in the presence of palmitate [51].

Furthermore, we investigated the phosphorylation and expression levels of mTOR and p70 S6K, in an attempt to elucidate the mechanisms involved in the effects of RSV. Our study found that exposure of L6 muscle cells to palmitate markedly increased the phosphorylation levels of mTOR and p70 S6K, and treatment with RSV blocked these effects (Figure 2, Figure 3 and Figure 8). These data are in agreement with previous studies showing increased phosphorylation/activation of mTOR and p70 S6K by palmitate in L6 [41] and C2C12 [52] muscle cells. In addition, skeletal muscle mTOR and p70 S6K phosphorylation/activation was increased in animal fed a high-fat diet [41,53]. Although we did not find any other studies examining the effects of RSV on skeletal muscle mTOR and p70 S6K, treatment of H9c2 cardiac myoblasts with RSV suppressed the high glucose and palmitate-induced mTOR and p70 S6K phosphorylation/activation [54]. In addition, our data showed an effect of RSV similar to metformin [55] and rapamycin [56]. Treatment of C2C12 muscle cells with metformin reduced the palmitate-induced mTOR and p70 S6K phosphorylation/activation [55]. Similarly, administration of rapamycin inhibited the high-fat diet-induced increase in mouse muscle mTOR and p70 S6K phosphorylation/activation [56]. Many studies showed that hyperphosphorylation/activation of mTOR and p70 S6K, results in increased serine phosphorylation of IRS-1 and contributes towards the development of insulin resistance [13,14,49]. A study found that a high-fat diet in rats significantly increased the phosphorylation/activation of muscle mTOR and p70 S6K and increased phosphorylation of ser^307^ and ser^636/639^ residues of IRS-1 [41]. Additionally, in p70 S6K-deficient mice, the ser^307^ and ser^636/639^ phosphorylation of IRS-1, induced by a high fat diet, was noticeably reduced, indicating that p70 6SK might be involved in the induction of insulin resistance [49]. Phosphorylation of the ser^307^ and ser^636/639^ residues of IRS-1 were shown to be catalyzed by mTOR and p70 S6K, resulting in reduced insulin signaling in muscle, adipose, and liver tissues [37,38,56]. Although our data showed an association between reduced serine phosphorylation of IRS-1 and reduced phosphorylation/activation of mTOR and p70 S6K by RSV treatment, further studies utilizing inhibitors or knockout approaches are required to prove that the reduced serine phosphorylation of IRS-1 was due to reduced mTOR and p70 S6K activation. In our previous study, we showed that short-term intralipid infusion resulted in a decrease of IκB alpha (marker of IKKβ activation) in muscle tissue, which was normalized by RSV [46]. Therefore, other kinases might be implicated in lipid-induced serine phosphorylation of IRS-1. Elevated inflammation levels were correlated with the development of muscle insulin resistance [57]. Exposure of C2C12 muscle cells to 100 µM RSV for 12 h resulted in reduction of all responses induced by palmitate treatment [58]. Tumor necrosis factor alpha (TNF-α) and interleukin-6 (IL-6) mRNA expression and protein secretion were reduced, and the increase in extracellular signal-regulated kinase (ERK), JNK, and IKKα/IKKβ phosphorylation, induced by palmitate, were completely abrogated by RSV treatment [58]. Furthermore, treatment with RSV (20 mg/kg b.w.) for 10 weeks, attenuated the increase in TNF-α, IL-1β, and IL-6 in muscles of diabetic rats [59]. Additionally, administration of RSV (150 mg/day for 30 days) attenuated the blood plasma inflammation markers (TNF-α and IL-6) in obese human subjects [60]. These studies indicate that inflammation plays a role in FFAs-induced insulin resistance and additional studies are required to elucidate the mechanism of action of RSV on inflammation markers and insulin resistance in muscle cells.

We also investigated the effects of RSV on the phosphorylation and expression levels of AMPK. Previous studies by us and others have found that treatment of L6 myotubes with RSV significantly increased the phosphorylation/activation of AMPK [28,32]. In the present study, we found that palmitate tended to increase the phosphorylation/activation of AMPK, but it did not reach significance. In contrast to our findings, exposure of L6 myotubes to palmitate, increased AMPK phosphorylation/activation [61,62]. These studies found that treatment of L6 myotubes with palmitate for 1 h, dose-dependently, increased the phosphorylation of AMPK with maximum stimulation at 400 µM palmitate, while the total levels of AMPK were unchanged [61,62]. The discrepancies between these studies and ours might therefore be due to the different time of treatment with palmitate. We exposed the cells to palmitate for 16 h, while in these studies exposure to palmitate was for only 1 h. It is possible that palmitate acutely increases AMPK activation and, over prolonged exposure, negative feedback control mechanisms are activated, which attempt to bring the AMPK activity levels back to the control basal levels.

In the present study treatment with RSV increased the phosphorylation/activation of AMPK. These finding are in agreement with other studies showing increased AMPK activation by RSV [10,32,63]. In contrast to the majority of the studies, Skrobuk et al., showed that treatment with 100 µM RSV for 4 h, inhibited the phosphorylation and activity of AMPK in primary human muscle cells [64]. The discrepancy between the study by Skrobuk and the other studies including ours is not clear and might be species-specific. Certainly, more studies utilizing human muscle cells should be performed in the future. Treatment with RSV significantly increased the phosphorylation/activation of AMPK, even in the presence of palmitate (Figure 1 and Figure 8). These data were in agreement with a previous study that showed that pre-treatment of endothelial cells with RSV increased AMPK phosphorylation/activation in the presence of palmitate [65]. Similarly, treatment with metformin resulted in increased phosphorylation/activation of AMPK in the presence of palmitate in C2C12 [55] and L6 [51] cells. Metformin is a plant-derived chemical and is widely used clinically as the first line of treatment for T2DM [66]. Metformin increases glucose uptake in muscle tissue by a mechanism that involves the activation of AMPK [67]. For these reasons, in the discussion of the present manuscript, we compared the action of RSV to metformin.

Studies have shown that AMPK activation directly lowers the activity of mTOR and p70 S6K by phosphorylation of the tuberous sclerosis complex 2 (TSC2) and raptor, leading to inhibition of its kinase activity [68], and AMPK can also inhibit IKKβ [46]. Our data suggest that the RSV-induced inhibition of mTOR and p70S6K might be mediated by AMPK. However, more studies should be performed in the future to investigate whether RSV has direct inhibitory effects on mTOR and p70S 6K, or whether the effects are mediated by AMPK.

Furthermore, we investigated the effect of RSV on glucose uptake, GLUT4 plasma membrane levels, and Akt phosphorylation/expression in palmitate-treated insulin-resistant L6 cells. Exposure of cells to palmitate alone had no effect on glucose uptake and plasma GLUT4 levels. Our findings are in agreement with other studies. Although, treatment of L6 [69] and C212 [70] muscle cells with palmitate-induced insulin resistance, did not affect the basal GLUT4 plasma membrane levels [69,70]. Treatment with RSV partially restored the insulin-stimulated glucose uptake (RSV+P+I: 138% ± 5.55% of basal control, *p* < 0.05; Figure 5), increased GLUT4 plasma membrane levels (RSV+P+I: 233% ± 1.8% of control, *p* < 0.01; Figure 6) and Akt phosphorylation/activation (RSV+P+I: 646% ± 101.6% of control, *p* < 0.05; Figure 7). The glucose uptake experiments were performed using the parental L6 cells whereas the experiments assessing the GLUT4 plasma membrane levels were performed using GLUT4myc overexpressing L6 cells. Therefore, the differences in the effects seen with RSV, namely the greater response of GLUT4 translocation, compared to the glucose uptake might be due to the different cells used. In our previous studies, incubation of GLUT4myc L6 myotubes with RSV for 24 h, increased the basal glucose uptake [28], similar to our current results with parental L6 myotubes. We recognize that glucose uptake should also be measured in the GLUT4myc cells in the future.

Our findings are the first to show that resveratrol might counteract the palmitate-induced muscle-insulin resistance. Overall, our data showed that in the presence of palmitate, treatment with resveratrol resulted in increased AMPK phosphorylation/activation, reduced palmitate-induced mTOR and p70 S6K activation, reduced serine phosphorylation of IRS-1, and increased insulin-stimulated Akt phosphorylation, GLUT4 plasma membrane levels, and glucose uptake (Figure 8).

Several studies examined the antidiabetic effects of resveratrol in vivo. In high-fat diet-induced diabetic animals, administration of resveratrol significantly reduced blood glucose and triglyceride levels, and improved insulin sensitivity [33,63,71]. However, the effects of resveratrol administration in individuals with metabolic syndrome are controversial. Despite a few studies that showed an improved metabolic profile [72,73,74], others found no significant improvement [75,76,77] and further long-term and well-controlled human clinical studies are required to have a good understanding of the effects of RSV in vivo.

## 4. Materials and Methods

### 4.1. Materials

Fetal bovine serum (FBS), resveratrol, o-phenylenediamine dihydrochloride (OPD), dimethyl sulfoxide (DMSO), cytochalasin B (CB), and anti-c-myc antibodies were purchased from Sigma Life Sciences (St. Louis, MO, USA). Minimum essential medium (α-MEM), trypsin, and antibiotic-antimycotic (100 U/mL penicillin, 100 µg/mL streptomycin and 250 ng/mL amphotericin B) were purchased from GIBCO Life Technologies (Burlington, Ont., Canada). Antibodies against total IRS-1, phospho-ser^307^ IRS-1, phospho-ser^636/639^ IRS-1, total mTOR, phospho-ser^2448^ mTOR, total p70-S6K, phospho-thr^389^ p70S6K, total AMPK, phospho-thr^172^ AMPK, total Akt, phospho-ser^473^ Akt, β-actin, and horseradish peroxidase (HRP)-conjugated anti-rabbit secondary antibodies were purchased from New England Biolabs (NEB; Mississauga, Ont., Canada). Additionally, peroxidase-conjugated goat anti-rabbit IgG were purchased from Jackson ImmunoResearch Lab (West Grove, PA, USA). [3H]-2-deoxy-D-glucose was purchased from PerkinElmer (Boston, MA, USA). Insulin (Humulin R) was from Eli Lilly (Indianapolis, IN, USA). Luminol Enhancer reagents, Bradford protein assay reagent, polyvinylidene difluoride (PVDF) membrane, and electrophoresis agents were purchased from BioRad (Hercules, CA, USA).

### 4.2. Palmitate Solution Preparation

Palmitate stock solution was prepared by conjugating fatty acids to free BSA, as previously reported [7]. In brief, palmitic acid was added in 0.1 N NaOH and diluted in 9.7% (w/v) BSA solution that was previously warmed to 45–50 °C. This solution gave a final stock concentration of 8 mM and a final molar ratio of free palmitate/BSA of 6:1.

### 4.3. Cell Culture and Glucose Uptake Assay

Parental and GLUT4myc overexpressing L6 skeletal muscle cells were grown in α-MEM media containing 2% v/v fetal bovine serum (FBS) and 1% (v/v) antibiotic-antimycotic solution, until fully differentiated. Myotube stage was reached roughly 6 to 7 days after seeding. All treatments were conducted in media containing 0% FBS. Fully differentiated myotubes were exposed to 0.4 mM palmitate, without or with 25 µM resveratrol for 16 h, followed by exposure to 100 nM insulin for 30 min. After treatment, the cells were washed with HEPES-buffered saline (HBS) solution and incubated with HBS containing 10-µM radiolabeled [3H]-2-deoxy-d-glucose for 10 min to measure glucose uptake. Non-specific glucose uptake was measured using 10 µM cytochalasin B. At the end of the 10 min, the cells were washed with 0.9% NaCl solution, followed by lysing with 0.05 M NaOH. The radioactivity was measured using liquid scintillation counter (PerkinElmer, Boston, MA, USA).

### 4.4. GLUT4myc Translocation Assay

L6 GLUT4myc overexpressing myotubes were grown and differentiated in 24-well plates. After treatment, the cells were washed with PBS and fixed using 3% paraformaldehyde-containing PBS for 10 min at 4 °C. The cells were then washed and exposed to 1% glycine containing PBS for 10 min at 4 °C, followed by incubation with a blocking buffer containing 10% goat serum in PBS for 15 min. The cells were then exposed to anti-myc antibody in blocking buffer for 60 min at 4 °C, followed by extensive washing with PBS and incubation with HRP-conjugated donkey anti-mouse IgG-containing blocking buffer for 45 min at 4 °C. The cells were extensively washed with PBS and incubated with O-phenylenediamine dihydrochloride (OPD) reagent, protected from light at room temperature, for 30 min. The reaction was stopped using 3N HCl solution. The supernatant was collected, and the absorbance measured at 492 nm. The OPD reagent was a substrate for HRP, therefore, produced a yellowish-orange color detected at 492 nm by an enzyme-linked immunosorbent assay (ELISA) plate reader (Synergy HT, BioTek Instruments, Winooski, VT, USA). The intensity of the color produced corresponded to the GLUT4myc transporters detected at the cell membrane.

### 4.5. Western Blotting

After treatment, the cells were washed twice with ice-cold HBS, followed by lysis buffer. The lysates were scraped off using a cell scraper and solubilized in 3x SDS sample buffer. Equal amount of protein sample (20 µg) was separated with sodium dodecyl sulfate polyacrylamide gel electrophoresis (SDS–PAGE) followed by transfer to a PVDF membrane. The membrane was then incubated with blocking buffer containing 5% w/v dry milk powder in Tris-buffered saline for 1 h, washed, and incubated overnight with primary antibody at 4 °C. The process was completed by exposure to HRP-conjugated anti-rabbit antibody for 1 h and Luminol Enhancer reagents (BioRad, Hercules, CA, USA). The bands were visualized using FluroChem software (Thermo Fisher, Waltham, MA, USA) and the densitometry was analyzed using ImageJ software. The cell lysates were stored at –20 °C.

### 4.6. Statistical Analysis

The data were analyzed using GraphPad Prism v7.0 (GraphPad Software, Inc. La Jolla, San Diego, CA, USA) and the significance of difference between the treatment groups was assessed using one-way ANOVA and Tukey’s post-hoc test. Statistical significance was presumed at *p* < 0.05.

## 5. Conclusions

Treatment of L6 skeletal muscle cells with palmitate, to mimic the elevated free fatty acid (FFA) levels often seen in obesity in vivo, induced insulin resistance. Exposure of the cells to palmitate significantly increased the serine phosphorylation of IRS-1 and mTOR and p70 S6K phosphorylation/activation, while the insulin-stimulated GLUT4 glucose transporter plasma membrane levels and glucose uptake were reduced. Treatment with RSV attenuated these palmitate-induced effects (Figure 8). RSV restored the insulin-stimulated glucose uptake, GLUT4 plasma membrane levels and Akt phosphorylation/expression in the presence of palmitate (Figure 8). In addition, RSV significantly phosphorylated/activated AMPK. Our study showed that RSV counteracted the FFA-induced muscle-cell insulin resistance. Further studies are required to fully elucidate the mechanism of RSV action and fully assess its potential as a therapeutic agent in the fight against insulin resistance and Type 2 diabetes mellitus.

## Figures and Tables

**Figure 1 ijms-21-04900-f001:**
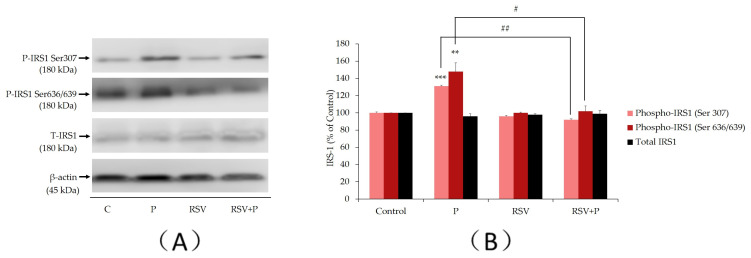
Effects of palmitate and resveratrol on serine (ser^307^ and ser^636/639^) phosphorylation and expression of IRS-1. Whole-cell lysates from L6 myotubes treated without (control, C) or with 0.4 mM palmitate (P), in the absence or the presence 25 µM resveratrol (RSV) were prepared, resolved by sodium dodecyl sulfate polyacrylamide gel electrophoresis (SDS–PAGE) and immunoblotted for phosphorylated ser^307^, ser^636/639^, total IRS-l or β-actin. A representative immunoblot is shown (**A**). The densitometry of the bands, expressed in arbitrary units, was measured using the Scion software. The data are the mean ± SE of three separate experiments presented as percent of control (**B**) (***p* < 0.01, ****p* < 0.001 vs. control; #*p* < 0.05, ##*p* < 0.01 as indicated).

**Figure 2 ijms-21-04900-f002:**
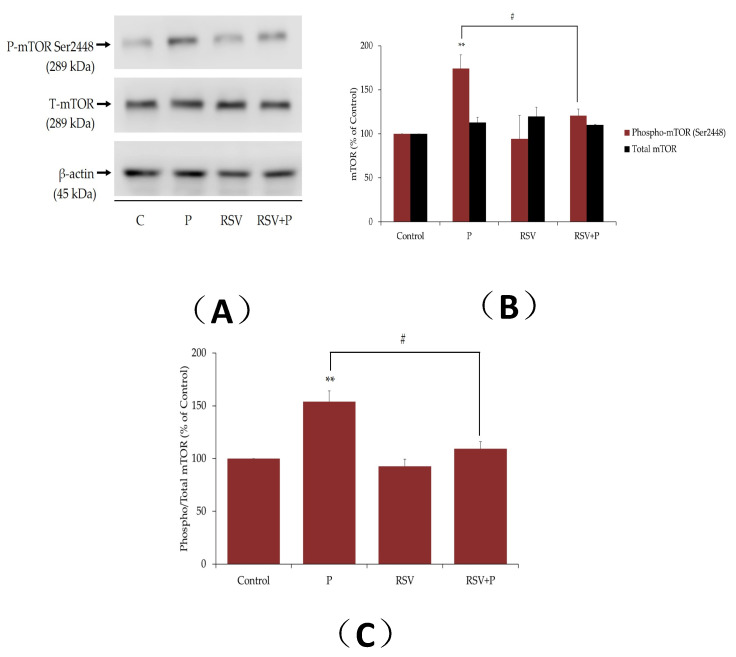
Effects of palmitate and resveratrol on mTOR phosphorylation and expression. Whole-cell lysates from L6 myotubes treated without (control, C) or with 0.4 mM palmitate (P) in the absence or the presence 25 µM resveratrol (RSV) were prepared, resolved by sodium dodecyl sulfate polyacrylamide gel electrophoresis (SDS–PAGE) and immunoblotted for phosphorylated ser^2448^, total mTOR, or β-actin. A representative immunoblot is shown (**A**). The densitometry of the bands, expressed in arbitrary units, was measured using the Scion software. The data are the mean ± SE of three separate experiments presented as percent of control (**B**). The data presented as the ratio of phosphorylated mTOR to total mTOR are shown (**C**) (***p* < 0.01 vs. control; #*p* < 0.05 as indicated).

**Figure 3 ijms-21-04900-f003:**
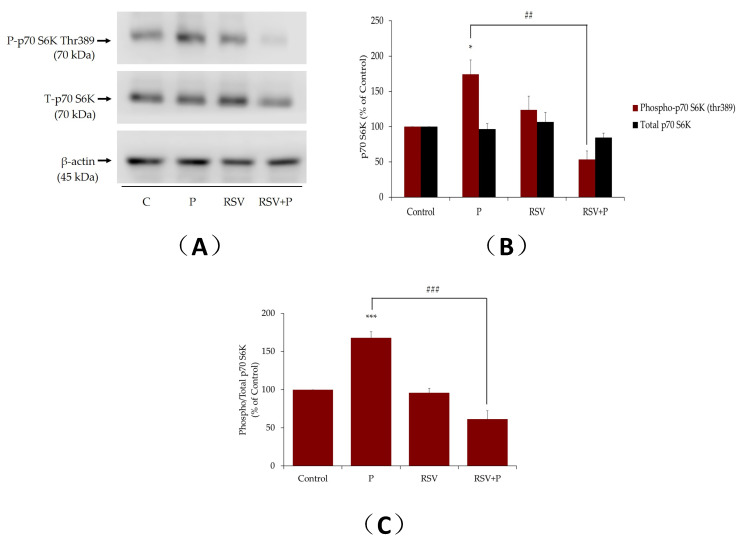
Effect of palmitate and resveratrol on p70 S6K phosphorylation and expression. Whole-cell lysates from L6 myotubes treated without (control, C) or with 0.4 mM palmitate (P) in the absence or presence of 25 µM resveratrol (RSV), were prepared, resolved by sodium dodecyl sulfate polyacrylamide gel electrophoresis (SDS–PAGE) and immunoblotted for phosphorylated thr^389^, total p70 S6K or β-actin. A representative immunoblot is shown (**A**). The densitometry of the bands, expressed in arbitrary units, was measured using the Scion software. The data are the mean ± SE of three separate experiments presented as percent of control (**B**). The data presented as the ratio of phosphorylated p70 S6K to total p70 S6K are shown (**C**) (**p* < 0.05, ****p* < 0.001 vs. control; ##*p* < 0.01, ###*p* < 0.001 as indicated).

**Figure 4 ijms-21-04900-f004:**
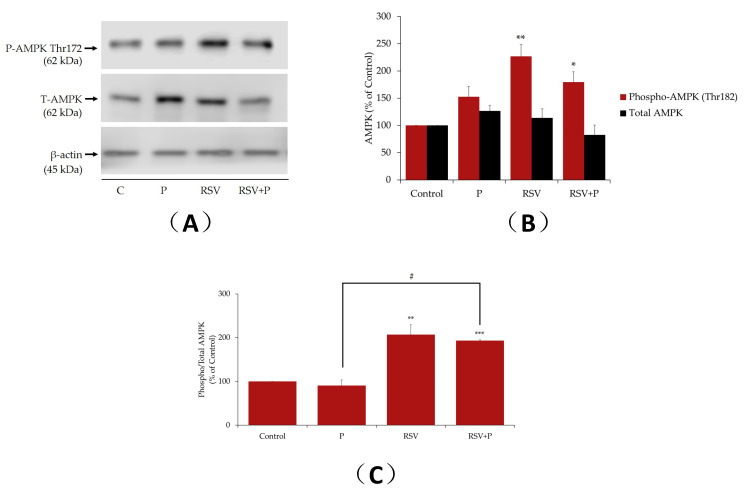
Effects of palmitate and resveratrol on AMPK phosphorylation and expression. Whole-cell lysates from L6 myotubes treated without (control, C) or with 0.4 mM palmitate (P) in the absence or the presence 25 µM resveratrol (RSV) were prepared, resolved by sodium dodecyl sulfate polyacrylamide gel electrophoresis (SDS–PAGE) and immunoblotted for phosphorylated thr^172^, total AMPK or β-actin. A representative immunoblot is shown (**A**). The densitometry of the bands, expressed in arbitrary units, was measured using the Scion software. The data are the mean ± SE of three separate experiments presented as percent of control (**B**). The data presented as the ratio of phosphorylated AMPK to total AMPK are shown (**C**) (**p* < 0.05, ***p* < 0.01, ****p* < 0.001 vs. control; #*p* < 0.05 as indicated).

**Figure 5 ijms-21-04900-f005:**
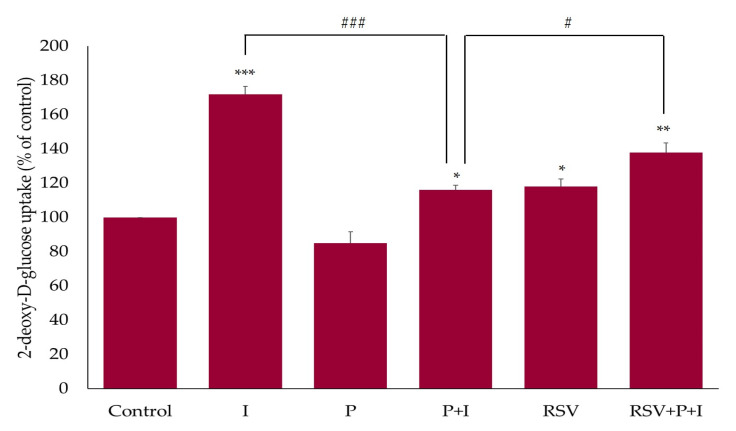
Effects of palmitate and resveratrol on insulin-stimulated glucose uptake. L6 myotubes were incubated without (control, C) or with 0.4 mM palmitate for 16 h (P), in the absence or presence of 25 µM resveratrol (RSV), followed by acute stimulation with 100 nM insulin for 30 min (I) and glucose uptake measurements. The values are the mean ± SE of three independent experiments, each performed in triplicates and expressed as percent of control (**p* < 0.05, ***p* < 0.01, ****p* < 0.001 vs. control; #*p* < 0.05, ###*p* < 0.001 as indicated).

**Figure 6 ijms-21-04900-f006:**
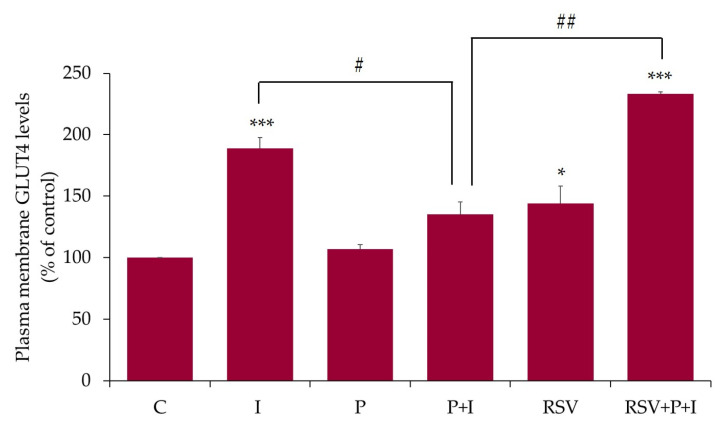
Effect of palmitate and resveratrol on GLUT4 translocation. GLUT4myc-overexpressing L6 myotubes were treated without (control, C) or with 0.4 mM palmitate for 16 h (P) in the absence or the presence of 25 µM resveratrol (RSV), followed by acute stimulation with 100 nM insulin for 30 min (I). After treatment, plasma membrane GLUT4 glucose transporter levels were measured. Results are the mean ± SE of three independent experiments performed in triplicates and expressed as percent of control (**p* < 0.05, ****p* < 0.001 vs. control; #*p* < 0.05, ##*p* < 0.01 as indicated).

**Figure 7 ijms-21-04900-f007:**
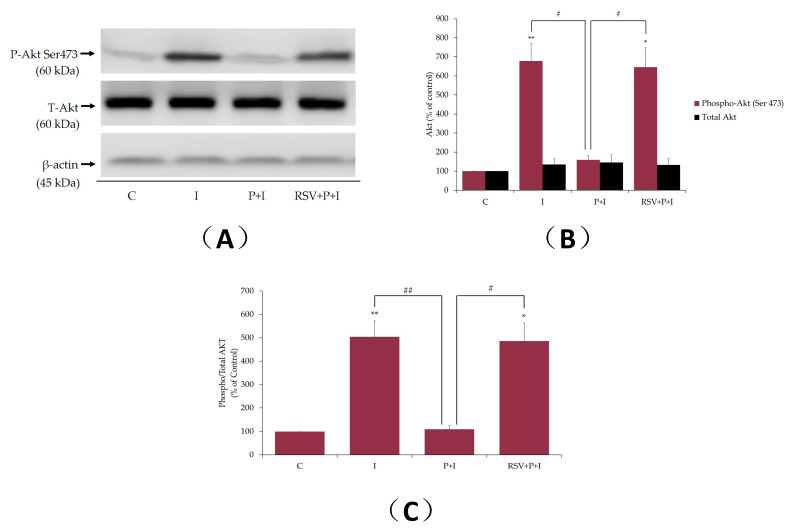
Effects of palmitate and resveratrol on insulin-stimulated Akt phosphorylation/expression. L6 myotubes were treated without (control, C) or with 0.4 mM palmitate for 16 h (P), in the absence or the presence of 25 µM resveratrol (RSV), followed by acute stimulation with 100 nM insulin for 30 min (I). Proteins (20 µg) in whole-cell lysates were resolved by sodium dodecyl sulfate polyacrylamide gel electrophoresis (SDS–PAGE) and immunoblotted for phosphorylated ser^473^Akt, total Akt, or β-actin. A representative immunoblot is shown (**A**). The densitometry of the bands, expressed in arbitrary units, was measured using the Scion software. The data are the mean ± SE of three separate experiments presented as percent of control (**B**). The data presented as the ratio of phosphorylated Akt to total Akt are shown (**C**) (**p* < 0.05, ***p* < 0.01 vs. control; # *p* < 0.05, ## *p* < 0.01 as indicated).

**Figure 8 ijms-21-04900-f008:**
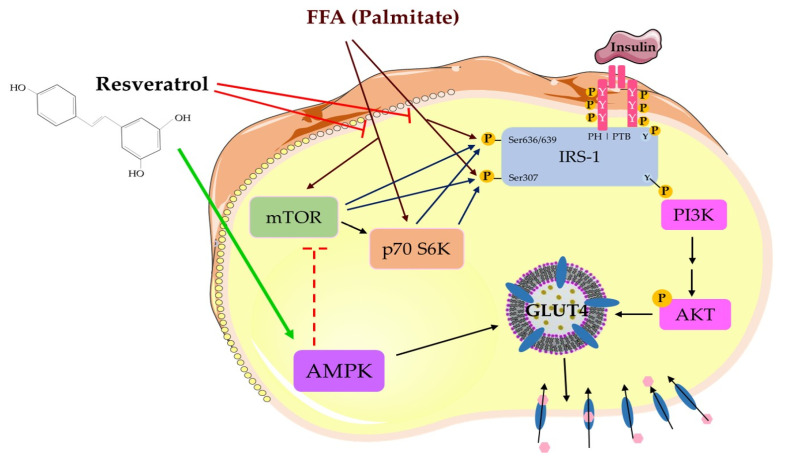
Resveratrol counteracted the free fatty acid (FFA)/palmitate-induced muscle cell insulin resistance. Resveratrol prevented the FFA-induced serine phosphorylation of IRS-1 and phosphorylation/activation of mTOR and p70 S6K, and increased the activation of AMPK.

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
