# Peer review of "Attenuation of Free Fatty Acid (FFA)-Induced Skeletal Muscle Cell Insulin Resistance by Resveratrol is Linked to Activation of AMPK and Inhibition of mTOR and p70 S6K"

_ijms, 2020, doi:10.3390/ijms21144900_

Round 1

Reviewer 1 Report

The study investigated how resveratrol affects skeletal muscle cell signalling and glucose uptake during palmitate treatment. While the experiments were carefully designed and executed, novelty of the findings were not apparent due to vast previous studies using similar methods and cell models.

Major comments:

  1. There were many reports that showed that resveratrol inhibited AMPK activity and glucose uptake in palmitate-treated skeletal muscle cells. One such example includes ‘Skrobuk, P., von Kraemer, S., Semenova, M.M. et al.Acute exposure to resveratrol inhibits AMPK activity in human skeletal muscle cells. Diabetologia 55, 3051–3060 (2012). https://doi.org/10.1007/s00125-012-2691-1’. The authors did not discuss any of these discrepancies.
  2. How did resveratrol affect insulin signalling: p-AKT, p-IRS-1 during insulin stimulation in your current study?
  3. There were too many citations (83) for an original article. May be worth removing the unnecessary ones.
  4. An introduction to resveratrol is missing in the Introduction. Moreover, the scientific question/objective/hypothesis was not clearly defined in the Introduction.
  5. Poor resolution images for all data figures, where the error bars were hard to be seen. Statistics was not stated in figure legends. One way or two-way ANOVA?
  6. Figure 6: How did palmitate alone affect GLUT4 translocation?
  7. Lines 214-221: This paragraph repeated what was in the Introduction. Consider to remove.
  8. Throughout the paper, the authors compared resveratrol to metformin, with which the reason was not clearly defined.
  9. The novelty of the study was not clearly defined. What did the current study add to the existing literature?

Minor comments:

  1. Line 25: Remove ‘(P)’
  2. Line 266: Remove the extra space at the beginning of the sentence.

Author Response

Please see the attachment, thanks.

Reviewer 2 Report

The study by Hartogh et al investigated changes in insulin signalling within L6 myotubes in response to palmitate and resveratrol treatment. Overall the paper is extremely well written, and the authors are to be congratulated on this point. The methodology and findings of the study are well described and discussed, and the authors have included a high number of citations to support their arguments (perhaps a few too many). However, I was left wanting a lot more from the results and most of my comments to the authors are along these lines.

Specific comments:

  • While western blots are routinely used to support other biological findings from in vivo/in vitro experiment, in the current manuscript there is an over-reliance on quantification of western blot data to infer changes in signaling proteins to justify cause-effect.
  • Why are all figures presented as percent change from control? Does the raw data vary highly from treatment to treatment?
  • Each figure states that 3-5 separate experiments were used to generate the figures. Can the authors be more specific and state in each figure legend what how many experiments were used for each group.
  • What control proteins are being used in the western blotting to ensure consistent sample loading? The authors should show representative blots including the protein ladder to ensure the bands shown are the correct/predicted size
  • Why are the protein phosphorylation events (figures 1-4) assessed only in the naïve state and in the absence of insulin? Whether RSV can modulate the insulin response in insulin resistant muscle is a key piece of information needed to support the authors conclusions.
  • Were Akt and PI3K changes assessed in response to palmitate and RSV?
  • Please present the glucose uptake data as a rate and not % change – for example, umol/min/kg or ug/kg/min as is the norm in the literature. Does RSV alone stimulate glucose uptake? Does RSV in combination with insulin alter glucose uptake? I was surprised not to see these groups included in the data. These groups should also be included in the glut 4 translocation data.
  • The authors need to comment on the discrepancy between GLUT 4 restoration (above insulin alone in figure 6) and the relatively smaller restoration in glucose uptake (well below the insulin alone in figure 5) in the RSV+P+I group.
  • Page 9, line 290 – the authors state that “Treatment with RSV completely restored the insulin-stimulated glucose uptake…”. This statement is not supported by the data which show that the glucose uptake in the RSV+P+I group appears to be markedly lower than the insulin alone group (figure 5). The authors need to soften their discussion on this point and provide the stats for this comparison in text or in the figure as appropriate.

Reviewer 3 Report

In this study, Den Hartogh et al, aim to identify the possible mechanism for resveratrol improvement of insulin resistance. In a model of L6 cell treated with palmitate they show resveratrol improve insulin sensitivity likely in an AMPK/mTOR- dependent pathway. The manuscript is well written and easy to follow. Nevertheless, some questions need to be addressed:

Major:

- The immunoblotting is lacking a specific control for the loading protein, i.e housekeeping protein. Without that appropriate control no conclusion can be reach.

- Insulin-induce glucose uptake seem to be only partially restored by resveratrol treatment (Figure 5). However, GLUT4 translocation to plasma membrane seem to be further increased in RVS+P+I when comparing with Insulin group (Figure 6). Have the authors any insights about this?

- Resveratrol have been found to reduce inflammation in muscle cells. Increase cytokine levels are linked to development of insulin resistance. Please briefly include this in the discussion of your proposed model.

-Figure 1 p-IRS1 Ser 636/639: WB images are not representatives for the graph. Specifically, it appears that there is a huge difference between Palmitate group and Palmitate+RVS, but no differences are shown in the graph. Please correct this inconsistency. Regarding this, as a suggestion, showing all groups together for each protein will facilitate the comparison between groups.

- Please show palmitate group in Figure 6. What is the reason not to show GLUT4 levels in that group?

Minor:

-Line 50: Did the authors mean mammalian target of rapamycin instead of mechanistic target of rapamycin (mTOR)?

Round 2

Reviewer 1 Report

No further concerns.

Author Response

We appreciate that this reviewer has No further concerns.

We think that our revised manuscript  is of high quality and deserves publication.

Reviewer 2 Report

Thank you to the authors for addressing most of the points raised. 

I have no further comments 

Author Response

(The authors gave the same response as above.)

Reviewer 3 Report

All questions have been addressed

Author Response

We are pleased that this reviewer is happy with all our answers and has No further concerns.

We think that our revised manuscript  is of high quality and deserves publication.